# AMOLED Pixel Circuit Using LTPO Technology Supporting Variable Frame Rate from 1 to 120 Hz for Portable Displays

**DOI:** 10.3390/mi13091505

**Published:** 2022-09-10

**Authors:** Ching-Lin Fan, Chun-Yuan Chen, Shih-Yang Liu, Wei-Yu Lin

**Affiliations:** 1Graduate Institute of Electro-Optical Engineering, National Taiwan University of Science and Technology, Taipei 10607, Taiwan; 2Department of Electronic and Computer Engineering, National Taiwan University of Science and Technology, Taipei 10607, Taiwan

**Keywords:** low-temperature polycrystalline silicon and oxide (LTPO), active-matrix organic light-emitting diode (AMOLED), low frame rate, high frame rate, variable frame rate, portable displays

## Abstract

This paper proposes a new 6T1C pixel circuit based on low-temperature polycrystalline oxide (LTPO) technology for portable active-matrix organic light-emitting diode (AMOLED) displays with variable refresh rates ranging from 1 to 120 Hz. The proposed circuit has a simple structure and is based on the design of sharing lines of switch-controlling signals. It also provides low-voltage driving and immunity to OLED degeneration issues. The calculation and analysis of programming time are discussed, and the optimal storage capacitor for the proposed circuit’s high-speed driving is selected. The results of the simulation reveal that threshold voltage variations in driving thin-film transistors of ±0.33 V can be well sensed and compensated with a 1.8% average shift of OLED currents in high-frame-rate operation (120 Hz), while the maximum variation in OLED currents within all gray levels is only 3.56 nA in low-frame-rate operation (1 Hz). As a result, the proposed 6T1C pixel circuit is a good candidate for use in portable AMOLED displays.

## 1. Introduction

Because of the benefits of outstanding color characteristics, short response time, high contrast ratio, and low power dissipation, active-matrix organic light-emitting diodes (AMOLEDs) have been widely used in display technologies [1,2,3]. The recent trend of display panels in high-end portable devices requires not only high resolution and pixels per inch but also low power consumption. Furthermore, portable products with adjustable frame rates are displayed with cutting-edge technology. High-frame-rate (HFR) technologies are used to achieve superior image quality in high-performance applications such as gaming or watching movies. In contrast, low-refresh-rate pixel designs are used in applications that require energy saving, such as smartwatches. In recent years, an increasing number of portable AMOLED displays have begun to incorporate LTPO technologies into pixel circuits. Low-temperature poly-Si (LTPS) thin-film transistors (TFTs) and metal-oxide-semiconductor (oxide) TFTs make up the LTPO structure. LTPS TFTs have been mainly adopted in mobile applications because of their high carrier mobility and excellent current driving capability, which make LTPS technologies beneficial to pixel circuit driving with low power consumption [4,5]. However, the characteristics of LTPS TFTs, such as the threshold voltage (V_TH_), would be significantly affected by the crystallization process [6,7]; additionally, due to the properties of polysilicon materials, LTPS TFTs generally suffer from high off-state leakage current [8,9]. Oxide TFTs, on the other hand, have advantages over LTPS TFTs due to their outstanding uniformity, extremely low off-leakage current, and low process complexity [10,11]. Both dynamic image quality and power consumption are important considerations for portable AMOLED displays, and using the LTPO structure is advantageous for achieving these two goals. Because of the high carrier mobility, driving the AMOLED pixel circuit with LTPS TFTs can dramatically reduce the programming period, allowing the pixel circuit to operate at HFR. Furthermore, to achieve low power consumption, the use of the LTPS TFT operates the pixel circuit with low-voltage driving due to its exceptional current driving capability; additionally, the low-frame-rate operation is a feasible way to realize power savings [12,13]. Holding the gate voltage of the driving TFT (DTFT) is required to support low-refresh rate driving; thus, oxide TFTs can be used to suppress the off-state current. Furthermore, the design of the storage capacitor is critical because the pixel circuit requires a relatively small capacitor to operate high-speed programming at HFR. 

As a result, numerous compensating pixel circuits based on LTPO have been published to alleviate DTFT threshold voltage variations. Apple Inc. [14] pioneered the LTPO concept with the 6T1C pixel circuit, which was mass-produced in the Apple Watch Series 4, enabling an adjustable frame rate between 1 and 60 Hz. Nevertheless, the control signal lines of the circuit were complicated, thus restricting the screen resolution and bezel area. Sharp Inc. [15] demonstrated a 7T1C pixel circuit with LTPO technology that drove an AMOLED panel favorably between 1 and 120 Hz using a GOA circuit. However, operating the circuit at 1 Hz consumed a significant amount of electricity because the emission control signal was still charged and discharged 120 times in one second. Following that, Fu et al. [16] proposed a 7T1C pixel circuit that successfully solved the frequent charging–discharging problem by holding the gate voltage of the LTPS driving TFT, allowing the circuit to drive between 1 and 120 Hz. Nonetheless, this 7T1C circuit is confronted with the IR-drop issue and intricate control signal line design.

This paper proposes a novel 6T1C pixel circuit with an LTPO structure for portable displays with variable frame rates ranging from 1 to 120 Hz. To reduce the number of control signal lines and TFTs in the proposed circuit structure, a holding period is used. In addition, the proposed circuit is independent of variations in the V_TH_ of OLED that provide immunity to OLED degeneration issues. The proposed circuit is provided with low-voltage driving of 5 V, operating at 120 Hz, with an average error rate of 1.8% of OLED currents, while the DTFT’s V_TH_ variations are ±0.33 V. Furthermore, although the proposed pixel circuit operates at 1 Hz, the overall variations in OLED driving currents are less than 3.56 nA. As a result, the proposed 6T1C pixel circuit is a good candidate for use in portable AMOLED displays.

## 2. The Operation of the Proposed Pixel Circuit

Figure 1a shows the circuit construction of the proposed 6T1C LTPO pixel circuit, which adopts the top-anode OLED configuration or the inverted top-emitting OLED (ITOLED) structure [17,18]. The proposed 6T1C pixel circuit is composed of one driving LTPS TFT (T1), two switching LTPS TFTs (T4 and T6), three switching oxide TFTs (T2, T3, and T5), and one storage capacitor (C_ST_). Furthermore, nodes N1 and N2 are the DTFT’s gate (V_G_) and source voltages (V_S_), which play an important role in controlling pixel currents. The corresponding timing diagram is shown in Figure 1b. To reduce the complexity of the pixel circuit, both Scan[n] and Scan[n − 1] control signals (Em[n] and Em[n − 1]) have the same pulse width. The operation of the proposed circuit is divided into four phases, as shown in Figure 2, presented in detail as follows.

### 2.1. Reset Stage

In the beginning stage of the operation, Scan[n − 1] and Em[n] are high to turn on T3 and T6. Scan[n] and Em[n − 1] are low to turn off T2, T5, and T4. The purpose of this stage is to reset the V_GS_ of the DTFT. The ELVDD is applied to N1 through T3, while node N2 is discharged to ELVSS. Furthermore, the OLED is completely turned off during the reset stage to prevent image flicker.

### 2.2. Programming Stage

During this stage, Scan[n − 1] and Em[n] are low to turn off T3 and T6. Em[n − 1] remains at a low voltage to prevent current from flowing through the OLED. The programming stage is intended to compensate for DTFT V_TH_ variations while simultaneously inputting data voltage signals. Scan[n] is raised to turn on T2 and T5, and a data voltage is applied to node N2 via T5. The diode-connected structure, which comprises T1, T2, and T5, discharges node N1 from ELVDD to V_DATA_ + V_TH_DTFT_. As a result, the voltage V_DATA_ + V_TH_DTFT_ is stored in C_ST_ at the end of the programming stage.

### 2.3. Holding Stage

Scan[n] becomes low during the holding period to turn off T2 and T5. To turn off T3 and T6, Scan[n − 1] and Em[n] are kept low. The N2 node is now floating, and the DTFT is turned off. As a result, even though Em[n − 1] turns on the switching T4, the OLED remains dark. Furthermore, the stored charges of C_ST_ are well held due to the low leakage current of oxide TFTs T2 and T3.

### 2.4. Emission Stage

In the final operating stage, Scan[n] and Scan[n − 1] remain low to turn off T2, T3, and T5, and Em[n − 1] remains high. Because the gate voltage (N1) is held at V_DATA_ + V_TH_DTFT_ and the ELVSS is applied to the source voltage (N2) while Em[n] becomes high to turn on T6, the driving TFT operates in the saturation region. The emitting OLED current can be calculated using the following equation:(1)$$\begin{array}{cc} I_{\mathrm{OLED}} & =\frac{1}{2}\mu_{n}C_{\mathrm{OX}}\;{\left(\frac{W}{L}\right)}_{\mathrm{DTFT}}\;{\left(V_{\mathrm{GS}}-V_{\mathrm{TH}\_\mathrm{DTFT}}\right)}^{2}\\ & =\frac{1}{2}\mu_{n}C_{\mathrm{OX}}\;{\left(\frac{W}{L}\right)}_{\mathrm{DTFT}}\;{\left(V_{\mathrm{DATA}}+V_{\mathrm{TH}\_\mathrm{DTFT}}-\mathrm{ELVSS}-V_{\mathrm{TH}\_\mathrm{DTFT}}\right)}^{2}\\ & =\frac{1}{2}\mu_{n}C_{\mathrm{OX}}\;{\left(\frac{W}{L}\right)}_{\mathrm{DTFT}}\;{\left(V_{\mathrm{DATA}}\right)}^{2}. \end{array}$$

Based on Equation (1), V_TH_DTFT_ is removed, so the OLED currents are solely determined by V_DATA_. As a result, the proposed pixel circuit not only compensates for variations in DTFT threshold voltage but is also unaffected by variations in ELVDD and V_TH_OLED_, improving image uniformity. In low-frame-rate applications, node N1 must be held at V_DATA_ + V_TH_DTFT_ to generate a constant driving current during the emission period. As a result, to reduce the voltage fluctuation of N1 caused by the leakage currents of T2 and T3, the switching TFTs connected to node N1 are implemented as oxide TFTs. In HFR applications, the proposed pixel circuit has to adopt an adequate storage capacitor to operate with high-speed discharging. Furthermore, the LTPS DTFT enables the proposed circuit to supply the OLED driving current with low-voltage power. As a result, the proposed pixel circuit can operate at rates ranging from 1 to 120 Hz with low-voltage driving, making it suitable for use in portable displays.

## 3. Analysis of Storage Capacitor

In HFR operation, using an appropriately sized storage capacitor to compensate for V_TH_DTFT_ variations is critical. To validate the storage capacitor design, the discharging process of the programming stage, as shown in Figure 3, is explained and analyzed in detail as follows.

In the equivalent circuit of the programming stage, the transient current (I_CST_) flowing through the storage capacitor can be expressed as
(2)$$I_{\mathrm{CST}}=(C_{\mathrm{ST}})\frac{{\mathrm{dV}}_{N1}}{\mathrm{dt}}$$
where V_N1_ is the voltage of N1 as well as the voltage being stored in C_ST_. In addition, as the Scan[n] signal turns on T2 and T5, a data voltage is supplied to N2 through T5. The diode-connected structure starts discharging N1 from ELVDD to V_DATA_ + V_TH_DTFT_ in an ideal operating situation. The discharging current (I_T1_) can be presented as
(3)$$I_{T1}=\frac{1}{2}\mu_{n}C_{\mathrm{OX}}\;{\left(\frac{W}{L}\right)}_{\mathrm{DTFT}}\;(V_{N1}-V_{\mathrm{DATA}}-V_{\mathrm{TH}\_\mathrm{DTFT}}{)}^{2}$$
where V_N1_ −V_DATA_ is the gate-to-source voltage (V_GS_) of the DTFT. Based on the principle of charge conservation, Equation (4) must be followed.I_CST_ + I_T1_ = 0.(4)

From Equations (2) and (3), Equation (4) can be derived as
(5)$$\frac{1}{V_{N1}-\;V_{\mathrm{DATA}}-\;V_{\mathrm{TH}\_\mathrm{DTFT}}\;}\;=\frac{1}{2}\mu_{n}C_{\mathrm{OX}}\;{\left(\frac{W}{L}\right)}_{\mathrm{DTFT}}\;(\frac{1}{C_{\mathrm{ST}}})\;\tau +A$$
where τ is the total time of the discharging process as well as the programming period in the proposed circuit, and A is the constant of integration. At the beginning of the discharging process, τ is equal to zero, and V_N1_ is ELVDD; thus, the constant of integration, A, can be calculated as
(6)$$A=\frac{1}{\mathrm{ELVDD}\;-{\;V}_{\mathrm{DATA}}-{\;V}_{\mathrm{TH}\_\mathrm{DTFT}}\;}$$

Further, at the end of the discharging process, τ is the total programming time, and V_N1_ is assumed to be discharged to β × (V_DATA_ + V_TH_DTFT_), where β ≥ 1, representing the actual voltage of V_N1_. Therefore, the formula for discharging period τ can be presented as
(7)$$\tau =\frac{2L}{\mu_{n}C_{\mathrm{OX}}W}\times C_{\mathrm{ST}}\times \left[\frac{1}{\beta \left(V_{\mathrm{DATA}}+{\;V}_{\mathrm{TH}\_\mathrm{DTFT}}\right)-{\;V}_{\mathrm{DATA}}-{\;V}_{\mathrm{TH}\_\mathrm{DTFT}}}-\frac{1}{\mathrm{ELVDD}\;-{\;V}_{\mathrm{DATA}}-{\;V}_{\mathrm{TH}\_\mathrm{DTFT}}}\right]$$

Equation (7) shows that as the mobility of the DTFT or V_TH_DTFT_ increases, the programming period can be reduced, which benefits high-refresh-rate driving. Furthermore, the storage capacitor C_ST_ is proportional to the programming time; thus, a relatively small C_ST_ is required for high-speed operation. To confirm the appropriate size of the storage capacitor used in the proposed 6T1C pixel circuit, a few LTPS driving TFT parameters and experimental data are required. The LTPS driving TFT had a geometry of 3 μm/10 μm, a threshold voltage of 1.5 V, mobility of 102 cm^2^/V·s, and a 30 nm SiO_2_ insulator C_OX_ of 115 nF/cm^2^. According to the simulation data, the parameter β was approximately equal to 1.0054, while V_DATA_ was −0.1 V when operating at the lowest gray level. Furthermore, to operate the proposed pixel circuit at 120 Hz with FHD resolution, the discharging time (data-inputting time) of the programming period must be less than the line-scanning time, which can be calculated as
(8)$$\mathrm{line}-\mathrm{scanning}\;\mathrm{time}=\frac{1\;s}{\mathrm{frame}\;\mathrm{rate}\;\times \;\mathrm{FHD}\;\mathrm{resolution}\;}$$
where “FHD resolution” represents the row number of the FHD resolution display, and the “frame rate” is 120 Hz. The discharging times of the proposed circuit have to be set below 7.716 μs for a frame rate of 120 Hz with an FHD resolution; as a result, the proposed storage capacitor C_ST_ must be limited as follows:(9)$$7.716\;\mu s\;>\frac{2L}{\mu_{n}C_{\mathrm{OX}}W}\times C_{\mathrm{ST}}\times \left[\frac{1}{\beta \left(V_{\mathrm{DATA}}+{\;V}_{\mathrm{TH}\_\mathrm{DTFT}}\right)-{\;V}_{\mathrm{DATA}}-{\;V}_{\mathrm{TH}\_\mathrm{DTFT}}}-\frac{1}{\mathrm{ELVDD}\;-{\;V}_{\mathrm{DATA}}-{\;V}_{\mathrm{TH}\_\mathrm{DTFT}}}\right]$$

In the calculation results of Equation (9), the storage capacitor C_ST_ needs to be lower than approximately 103 fF. As a result, we set 100 fF as the value of the proposed C_ST_.

## 4. Results and Discussion

LTPS and a-IZTO TFT devices were fabricated to validate the performance of the proposed 6T1C pixel circuit, and the measured and simulated transfer curves are shown in Figure 4. The a-IZTO TFTs, using a bottom-gate structure, can achieve a low leakage current. The LTPS TFTs, using a top-gate structure, are able to provide high carrier mobility and excellent current driving capability. The a-IZTO and LTPS TFTs used in the proposed pixel circuit were measured by a semiconductor parameter analyzer (HP4145B) to obtain the transfer curves. Furthermore, the characteristics of the a-IZTO and LTPS TFTs were fitted by software to acquire the model parameters before the circuit simulation. The proposed circuit was simulated using AIM-Spice, and the parameters are listed in Table 1. Herein, the storage capacitor C_ST_ is set to 0.1 pF to operate the proposed pixel circuit at an HFR of 120 Hz with FHD resolution. Furthermore, designing the Scan and Em with the same voltage range reduces the complexity of GOA and is advantageous for narrow-bezel applications.

Figure 5a shows the simulated transient waveforms of node N1 of the proposed pixel circuit operating at 120 Hz when V_TH_ variations in DTFT are −0.33, 0, and +0.33 V. Node N1 is reset to ELVDD (5 V) at the start of the operation and then discharged to approximately V_DATA_ + V_TH_ (3 V) with an input data voltage of 1.5 V after the programming period. Furthermore, the proposed circuit senses variations of 0.33 and 0.32 V, which are nearly equal to the actual variation of 0.33 V. Furthermore, at the end of the holding period, node N1 drops from 3 to 2.85 V due to the parasitic capacitance issue caused by clock-feedthrough effects. Figure 5b shows the simulated OLED currents versus data voltage from −0.1 to 1.9 V for ±0.33 V DTFT threshold voltage shifts with an average error rate of 1.8%, demonstrating the proposed pixel circuit’s high compensating capability for threshold voltage variations when driven at 120 Hz. 

Figure 6 depicts the simulated voltage waveforms of node N1 operating at a low frame rate of 1 Hz when the V_TH_ variations in the driving TFT are −0.33, 0, and +0.33 V. After the reset, programming, and holding operations, the voltage of node N1 ought to be sustained within the long period emission stage of 1 s. It can be seen that the voltage of node N1 is fine and holds that the maximum variation is about 0.23%. Furthermore, at the end of the emission period, the proposed circuit detects voltages of 0.328 V and 0.326 V, indicating that the proposed pixel circuit is highly resistant to DTFT threshold voltage variation when driven at a low frame rate of 1 Hz.

Figure 7a depicts the simulated OLED currents at the low gray level in the proposed circuit structure using all-LTPS TFTs or LTPO for a 1 s emission period. Within the long emission time, being lower than the ELVDD, the voltage of node N1 (V_DATA_ + V_TH_DTFT_) is mainly charged by the path of the leakage current of T3. As a result, the OLED current increases dramatically as the voltage of node N1 rises. Because of the extremely low off-current characteristic of oxide TFTs, leakage currents can be limited in LTPO applications. Figure 7b depicts the simulated OLED currents emitting continuously for 1 s at various gray levels while the proposed 6T1C pixel circuit operates at a low frame rate of 1 Hz. The overall variations in OLED driving currents are less than 3.56 nA, indicating excellent stability when operating at a low frame rate of 1 Hz. 

In addition, in Table 2, a comparison between the proposed and previously published pixel circuits demonstrates the advantages of the proposed 6T1C circuit, including a simple structure, a minimum number of signal lines, wide refresh rate support, and low voltage driving. The total number of signal lines, in this case, includes ELVDD, ELVSS, V_DATA_, V_REF_, and switch-controlling signal lines. Finally, based on the simulated results, it is confirmed that the proposed 6T1C pixel circuit has good performance when operating at various refresh rates ranging from 1 to 120 Hz, making it suitable for portable displays.

The layout of the proposed pixel circuit using the method of line sharing is shown in Figure 8. The target display specifications in this paper are designed for portable applications, including full high-definition (FHD) resolution, a high ppi of 525, and 1 Hz to 120 Hz frame rate support. With the small subpixel dimension of 48.5 µm × 24.25 µm, it is believed to reach a high pixel density for portable applications.

## 5. Conclusions

This paper proposes a new 6T1C pixel circuit for AMOLED portable displays based on LTPO technology that supports variable frame rates ranging from 1 to 120 Hz. To use it in narrow-bezel applications, identical switch-controlling signals can be used to reduce circuit complexity.

Furthermore, the proposed circuit could not only be driven with a low power voltage of 5 V but also be free of the OLED degeneration problem. According to the results of the calculation and analysis, the storage capacitor of the proposed circuit, being suitable for HFR driving, is chosen as 100 fF. The results of the simulation show that the average error rate of OLED currents with ±0.33 V threshold voltage variations in the DTFT is 1.8% in the HFR driving scheme (120 Hz); additionally, during low-frame-rate operation (1 Hz), the maximum shift of OLED currents at all gray levels is about 3.56 nA. As a result, the proposed pixel circuit can be successfully operated at a variety of refresh rates ranging from 1 to 120 Hz, which is advantageous for portable AMOLED displays.

## Figures and Tables

**Figure 1 micromachines-13-01505-f001:**
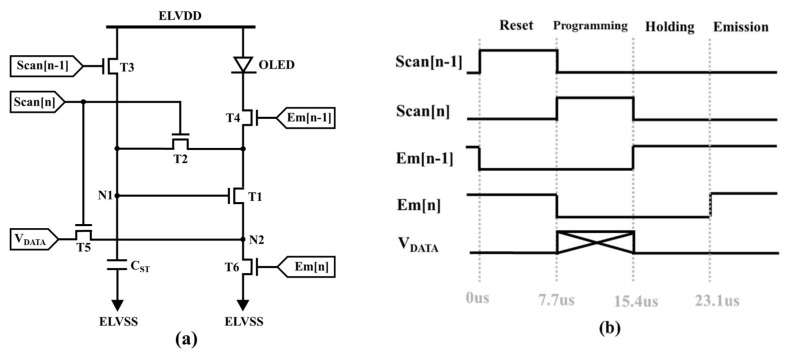
(**a**) The schematic of the proposed 6T1C pixel circuit and (**b**) the timing diagram of the control signals.

**Figure 2 micromachines-13-01505-f002:**
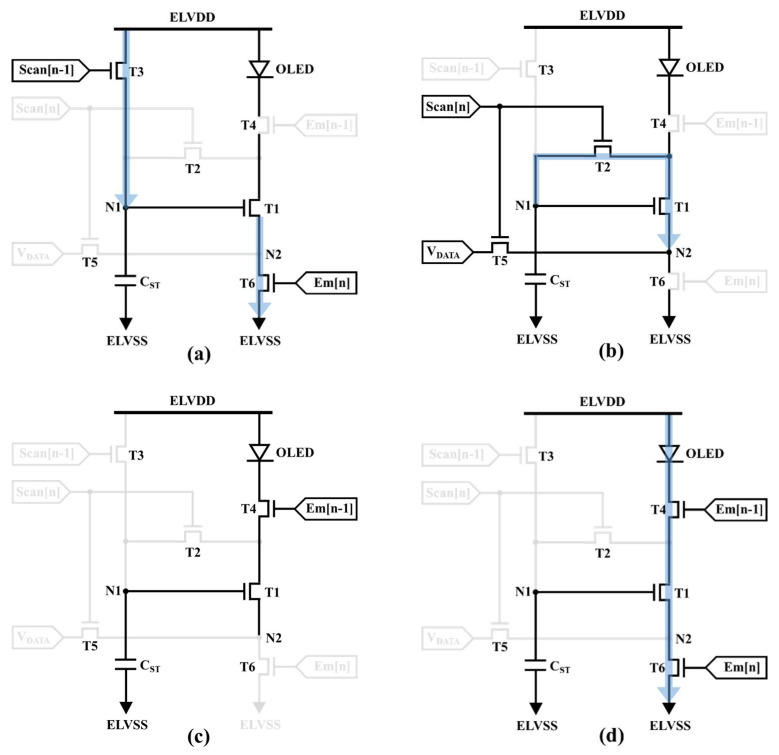
Schematic of pixel circuit operation, including (**a**) reset, (**b**) programming, (**c**) holding, and (**d**) emission periods.

**Figure 3 micromachines-13-01505-f003:**
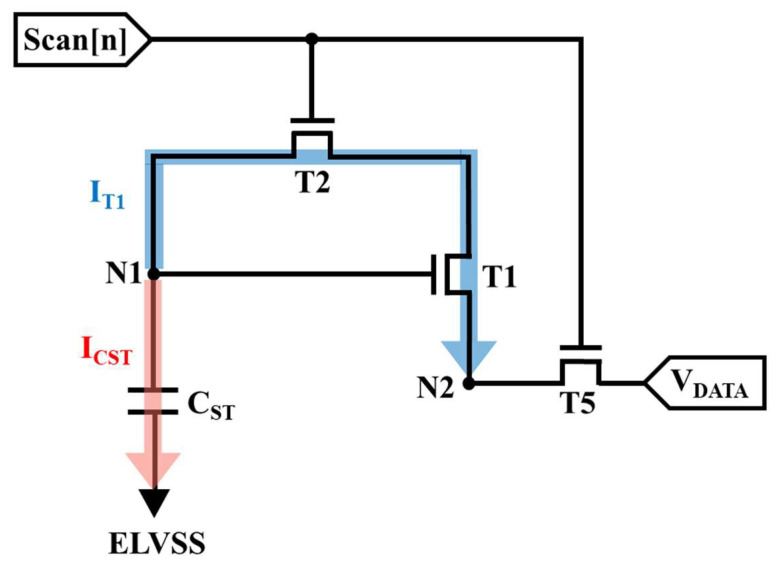
Equivalent circuit of programming stage for the calculation of discharging time.

**Figure 4 micromachines-13-01505-f004:**
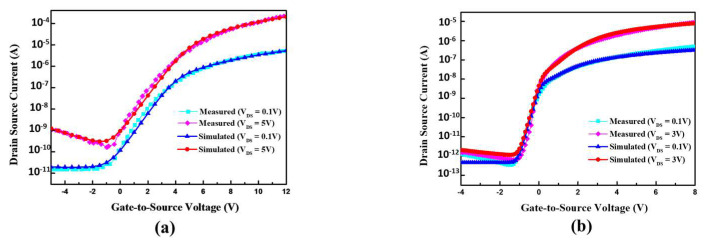
Measured and simulated (**a**) LTPS TFT and (**b**) a-IZTO TFT transfer curves.

**Figure 5 micromachines-13-01505-f005:**
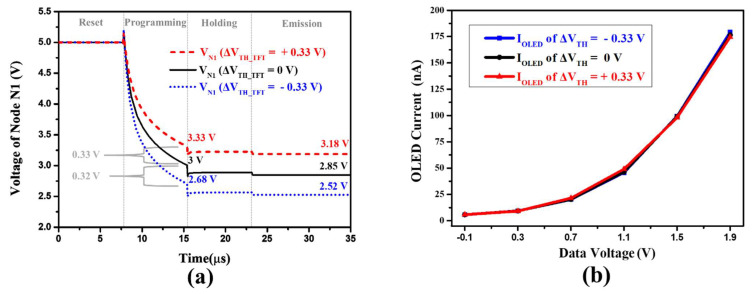
The (**a**) simulated transient waveforms of node N1 voltage operating at 120 Hz when V_TH_ variations in DTFT are −0.33, 0, and +0.33 V. (**b**) Simulated OLED driving currents versus data voltage of the proposed 6T1C pixel circuit for ±0.33 V threshold voltage shifts of DTFT.

**Figure 6 micromachines-13-01505-f006:**
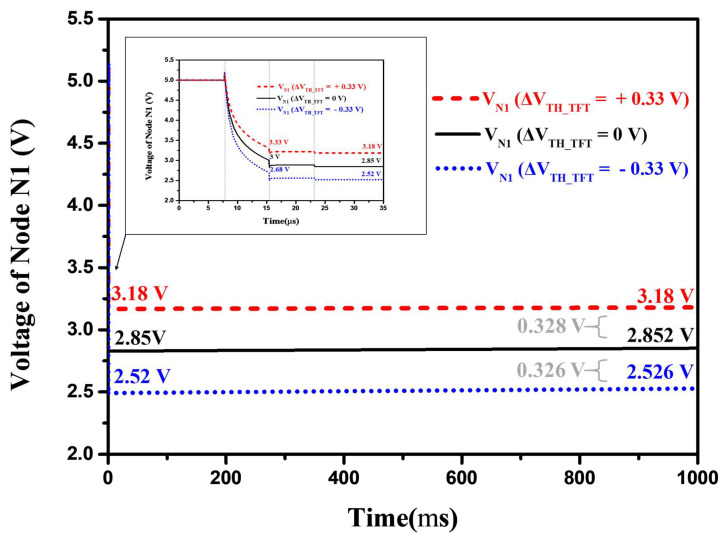
Simulated waveforms of the voltage of node N1 operating at 1 Hz when V_TH_ variations in DTFT are −0.33 V, 0 V, and +0.33 V.

**Figure 7 micromachines-13-01505-f007:**
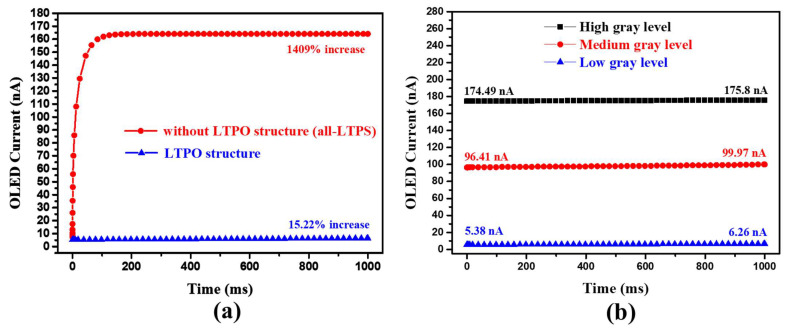
(**a**) Simulated OLED currents with a long emission time of 1 s using all-LTPS TFTs or LTPO structure for the proposed pixel circuit. (**b**) Simulated OLED currents with a long emission time of 1 s at low, medium, and high gray levels for the proposed 6T1C pixel circuit.

**Figure 8 micromachines-13-01505-f008:**
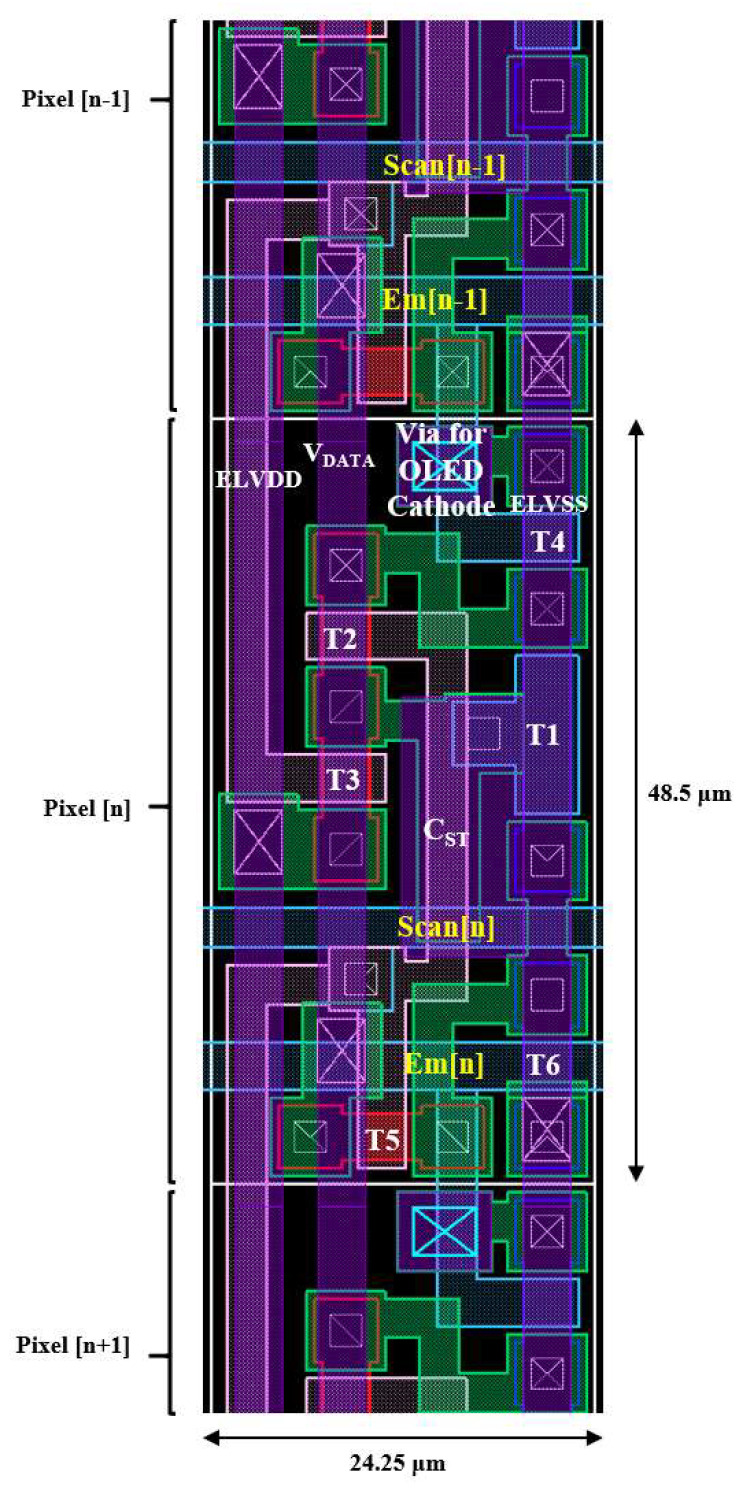
The layout of the proposed 6T1C pixel circuit.

**Table 1 micromachines-13-01505-t001:** Specifications of proposed 6T1C pixel circuit.

Parameter	Value	Parameter	Value
(W/L)_T1_ (μm)	3/10	ELVDD (V)	5
(W/L)_T2−T6_ (μm)	3/3	ELVSS (V)	0
C_ST_ (pF)	0.1	Scan, Em (V)	−1~5
C_OLED_ (pF)	0.4	V_DATA_ (V)	−0.1~1.9
Emission period for 120 Hz (ms)		8.33	
Emission period for 1 Hz (s)		1	

**Table 2 micromachines-13-01505-t002:** Comparison between proposed and previously published pixel circuits.

Reference	This Study	Ref. [14]	Ref. [16]	Ref. [19]	Ref. [20]
Structure	6T1C	6T1C	7T1C	6T2C	7T1C
Total signal lines	5	8	7	7	6
Frame rate	1–120 Hz	1–60 Hz	1–120 Hz	120 Hz	15–60 Hz
Resolution	1920 × 1080	368 × 448	1920 × 1080	1920 × 1080	320 × 360
Power voltage (V_DD_ − V_SS_)	5 V	N/A	7 V	4.8 V	6.6 V

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
