# Peer review of "AMOLED Pixel Circuit Using LTPO Technology Supporting Variable Frame Rate from 1 to 120 Hz for Portable Displays"

_micromachines, 2022, doi:10.3390/mi13091505_

Round 1

Reviewer 1 Report

Fan et al. report the LTPO-based 6T1C pixel circuit with shared switching-controlling signal lines. Both calculations and experiments were conducted to verify their proposal. The results of the simulation reveal that the delta Vt of driving TFTs of ±0.33 V can be well sensed and compensated with the 1.78% average shift of OLED currents at 120 Hz, while the maximum variation of OLED currents within all gray level is only 3.56 nA at 1 Hz. The results are quite interesting, thus I would recommend publishing in Micromachines after major revision.

1. The font of labels (a) and (b) in Figure 1 is not the same.

2. The authors should provide the details as much as possible for the TFT simulation by using AIM-Spice, also the fabrication details for these TFTs.

4. No experimental section is given in the paper, and the details of making 6T1C circuit are totally missing. 

5. The optical and SEM images of 6T1C unit should be provided.

Reviewer 2 Report

The submitted paper seems out of the scope of the journal, Micromachines. The followings must be clearly responded.

1. In the proposed circuit, the anode of the OLED is not directly connected to source/drain of TFT (M4). ITO is used for the anode of the OLED via sputtering process. The plasma process is very harmful to organic layers of OLED. Thus, ITO must be formed before depositing organic layers. In this reason, the anode of OLED is generally connected to source/drain of TFTs in 6T1C or 7T1C pixel circuits. The OLED connection shown in Fig. 1(a) seems strange. Please explain why the anode of OLED is not connected to source/drain of TFTs. Please describe any reason if the authors have any solution on the process issue caused by the given circuit connection.

2. Table 2 shows the comparison results. The number of the total signal lines of the proposed circuit is 5. This means ELVDD and ELVSS are not counted as signal lines. Then, this must be applied to other circuits fairly. The circuit of [14] has 6 signal lines if ELVDD and ELVSS are not included. Please correct Table 2 with a fair comparison.

3. The author claimed that IR-drop issue can be solved. However, IR-rise issue can be induced by ELVSS line. Please explain why there is no issue on IR-rise from ELVSS.

4. Fig. 4(b) shows the transfer characteristics of oxide TFT. But, the leakage current was the order of 1 pA or higher. CST was only 0.1 pF as shown in Table 1. If the leakage current flows in the order of 1 pA via T3 or T2 for 1 second, then the voltage change of CST becomes 10 V. This means 1 Hz operation of the circuit is impossible. The voltage change across CST must be simulated. But, the simulation results seems different from the simple calculation. The authors must describe the leakage model of oxide TFTs. It would be helpful if the authors show the voltage change across CST by showing the currents through T3 or T2.

5. Table 1 shows COLED is 0.1 pF. Could you explain why COLED is 0.1 pF? Generally, COLED is higher than 0.1 pF the authors presented because the organic layers are very thin. Please specify why the authors assumed COLED of 0.1 pF.

Round 2

Reviewer 1 Report

The authors have addressed my concerns, and it can be accepted now.

Reviewer 2 Report

1.      Authors claim that there is no problem in their pixel circuit because top-anode OLED and inverted top-emitting OLED can be used. In addition, they claim that the pixel structure based on top-anode OLED have been widely used in AMOLED pixel circuits. However, either top-anode OLED or ITOLED is not widely used in commercially available OLED displays. Only some researchers proposed the AMOLED pixel structures using top-anode OLED or ITOLED. It is not right to generalize the usage of top-anode OLED or ITOLED in mobile OLED displays. Authors need to limit their research scope within mobile displays adopting top-anode OLED or ITOLED structure. Please include comments that the research is limited to AMOLED pixel circuits adopting top-anode OLED or ITOLED structure.

2.    I don’t agree with the authors’ response to the IR-rise issue. The authors claimed that the voltage of node N1 is discharged from ELVDD to VDATA + VTH_DTFT + ΔVSS above the ground (ELVSS). It’s wrong! The N1 voltage will not become VDATA + VTH_DTFT + ΔVSS but VDATA + VTH_DTFT. The current path does not include ELVSS line. The current stops when VGS of the driving TFT becomes VTH_DTFT, which means that the N1 voltage becomes VDATA + VTH_DTFT. It is impossible to include IR-rise voltage of ELVSS in N1 node. Please do not claim that the proposed pixel circuit is immune to IR-drop or IR-rise in the revised manuscript.

3.      The authors showed that the leakage currents of T2 and T3 were 0.00637 pA and 0.011 pA, respectively. Then, the total leakage current affecting the N1 voltage was 0.01747 pA. Thus, the voltage change of the node N1 in one second becomes 0.1747 V. But, the simulation results were different as shown in Fig. 6 of the revised manuscript. The biggest voltage change was 0.006 V as shown in Fig. 6. The calculated voltage change based on the leakage currents provided by the authors did not match the simulated voltage change. Please explain the mismatch, 0.01747 V vs. 0.006 V.

4.      The smaller area makes the COLED smaller. I agree with it. However, the thickness of OLED cell between the anode and cathode was not explained by the authors. 7T1C pixel circuit was developed to discharge the charge stored in COLED. If COLED is small as the authors claim, the 7T1C pixel circuit must not be developed. The authors response to COLED has too weak logical base.

Round 3

Reviewer 2 Report

Most comments are appropriately responded.

But Response 4 needs more explanation. Is the OLED device in reference [1] by C.-J. Shih the same as the OLED used in commercially available OLED displays? So it's not fair to refer the C-V. In addition, the capacitance of the OLED device becomes the highest as current increases, which means its capacitance below Vt is smaller than that above Vt. The impedance of OLED comprises parallel connection of resistor and capacitor. In my opinion, the C-V curve is not correct.